# Information Overload, Wellbeing and COVID-19: A Survey in China

**DOI:** 10.3390/bs11050062

**Published:** 2021-04-27

**Authors:** Jialin Fan, Andrew P. Smith

**Affiliations:** 1School of Psychology, Shenzhen University, Shenzhen 518000, China; FanJL@szu.edu.cn; 2School of Psychology, Cardiff University, Cardiff CF10 3AS, UK

**Keywords:** COVID-19, China, information overload, wellbeing

## Abstract

(1) Psychology must play an important role in the prevention and management of the COVID-19 pandemic. The aim of the present study was to examine associations between the perceptions of information overload and wellbeing in China during the initial phase of COVID-19. (2) Methods: The present research involved a cross-sectional online survey, which controlled for established predictors of wellbeing and the perception of general (not COVID-19-specific) information overload. The setting of the research was China, February 2020. A total of 1349 participants completed an online survey, and the results from 1240 members of the general public who stated that they were uninfected are reported here (55.6% female; 49.4% single; age distribution: 17–25 years: 26%; 26–30 years: 24.3%; 31–40 years: 23.9%; 41–50 years: 16.2%; 51 years+: 9.6%; the most frequent occupations were: 21.5% students; 19.5% teachers; 25.9% office workers; 10.8% managers, plus a few in a wide range of jobs). The outcomes were positive wellbeing (positive affect and life satisfaction) and negative wellbeing (stress, negative affect, anxiety and depression). (3) Results: Regressions were carried out, controlling for established predictors of wellbeing (psychological capital, general information overload, positive and negative coping). Spending time getting information about COVID-19 was associated with more positive wellbeing. In contrast, perceptions of COVID-19 information overload and feeling panic due to COVID-19 were associated with more negative wellbeing. (4) Conclusions: These results have implications for the communication of information about COVID-19 to the general public and form the basis for further research on the topic.

## 1. Introduction

Psychological characteristics and behaviour will play a key role in the COVID-19 pandemic [1]. Prevention of infection requires appropriate hygiene [2] and social distancing [3]. Social isolation will reduce exposure to the virus, but there is also evidence of increased stress and reduced wellbeing during quarantine [4]. Indeed, reports from the media in the UK show that lockdown has led to increased social conflict and abuse. Management of COVID-19 by healthcare professionals requires the wearing of appropriate protective equipment, following appropriate procedures and coping with death and dying. It is not surprising, therefore, that healthcare professionals in China who have had to care for COVID-19 patients reported an increase in mental health problems [5]. In the general population, COVID-19 has health consequences beyond the direct effects of the virus, with reduced mental wellbeing and increased psychological distress being widely reported [6,7,8,9,10,11,12]. 

### 1.1. Nature and Extent of Communication about COVID-19 

An additional risk factor for mental health problems has been the nature and extent of communication about the pandemic. Initially, it was recommended to increase information and communication technology to reduce anxiety and social isolation [13,14,15]. However, this increase led to greater problematic internet use in China [16], and social media exposure was associated with greater anxiety and depression there [17]. As well as problems due to the amount of information about COVID-19, there have been issues related to the accuracy of the news [18,19]. Indeed, it is important to consider the direct effects of the COVID-19 pandemic, the effects of prevention and management strategies on mental wellbeing and the “infodemic” [20], which also helps to create the “perfect storm” that affects physical and mental health.

The aim of the present study was to examine associations between perceptions of information overload and wellbeing in China in the first wave of COVID-19. Information overload (IO) is the state of stress experienced when the amount of information given exceeds the limit of the user’s information processing capacity [21]. Excessive information about COVID-19 may lead to information overload, which could have a negative effect on wellbeing. Information overload from other sources and wellbeing have been investigated in several studies [22,23,24,25,26]. Findings confirm the negative effect of information overload on wellbeing, although two studies demonstrated a positive effect if the person’s use is controlled. There are many causes of information overload, and a questionnaire, the Perceived Information Overload Scale (PIOS; [27]), has been developed to measure exposure to these. 

### 1.2. Demands–Resources–Individual Effects and the Wellbeing Process

Wellbeing is a difficult concept to define and involves many different factors. The “wellbeing process model” [28] was used as the theoretical framework here, and it provides a holistic approach to wellbeing and a measuring instrument that is useful in practice and policy. The “wellbeing process model” was based on the “Demands–Resources–Individual Effects (DRIVE) model”, which was used to advance research in occupational stress [29]. The DRIVE model (shown in Figure 1) included job characteristics, perceived stress, personal characteristics such as coping styles and negative outcomes such as anxiety and depression.

Initial research using the DRIVE model [30,31] found strong support for direct effects of the predictors but little evidence of moderation or mediation. The next development of the DRIVE model [32,33] was to include positive personality characteristics such as psychological capital (self-esteem, self-efficacy and optimism) and positive appraisals (e.g., job satisfaction) and outcomes (e.g., happiness and positive affect). Positive outcomes form the basis of most approaches to subjective wellbeing, but it is important to include both positive and negative aspects of wellbeing as they involve different CNS mechanisms. The Wellbeing Process Questionnaire involved initial development that demonstrated significant correlations between short measures and the longer scales from which they were derived [34]. Research [35,36,37,38] using the Wellbeing Process Questionnaire (WPQ) has identified established predictors of positive and negative wellbeing, with positive factors such as psychological capital being associated with positive outcomes and negative factors such as information overload and negative coping styles associated with negative outcomes. Again, the results suggest independent, additive effects rather than interactions between variables. The general principles of the wellbeing process model are shown in Figure 2.

### 1.3. The Present Study

After the initial development of the wellbeing process model, research examined whether new variables (e.g., fatigue, rumination, daytime sleepiness) added to the predictive power when the established predictors were statistically controlled. This approach was continued here, with interest being in the associations between COVID-19-related information overload and positive and negative wellbeing outcomes. The analyses controlled for demographic factors and the established predictors of wellbeing outcomes. The established predictors from the wellbeing process model were general information overload (GENIO), negative coping (NEGCOP), psychological capital (PSYCAP) and positive coping (POSCOP). These predictor variables usually have independent effects, and prior research has shown little evidence of interactions between the variables. Measurement of these variables allowed testing of the hypothesis that COVID-19-related information overload would have associations with positive and negative wellbeing outcomes that were independent of the established predictors. 

GENIO was measured using items from the Perceived Information Overload Scale [27]. The Perceived Information Overload Scale has good internal consistency (α = 0.86) and validity. The scale consists of 16 items that measure two subscales of information overload, environment-based and cyber-based information overload. Although information overload is an indicator of stress, results indicate that the Perceived Information Overload Scale score and the Perceived Stress Scale score do not overlap, which suggests that cyber-based and place-based information overload scales measure different concepts from perceived stress [27]. The present study adapted the PIOS so that items were related to communication about COVID-19 or to more general aspects of information overload. Examples of these items are shown in the Methods section. 

The coping measures were derived from the Revised Ways of Coping Checklist [39]. The negative coping measure (NEGCOP) included avoidance, self-blame and wishful thinking (α = 0.85). The positive coping measure (POSCOP) included problem-focused coping and seeking social support (α = 0.82). The measure of psychological capital (PSYCAP; α = 0.90) included measures of self-efficacy [40], self-esteem [41] and optimism [42].

The COVID-19-related variables covered information overload about COVID-19 (COVID-IO), measured by adapting items from the perceived information overload scale to the COVID-19 context. The amount of time spent getting information about COVID-19 (COVID-TIME) from the media was also measured. The amount of attention paid to COVID-19 (COVID-ATT), issues related to the use of masks and hand sanitisers (MASKS) and panic due to COVID-19 (PANIC) were also recorded. Examples of these items are shown in the Methods section.

### 1.4. Objectives and Hypotheses

The aims and objectives of the present study were to use a multivariate approach, based on information overload and wellbeing models, to examine associations between overload from COVID-19 communications and positive and negative wellbeing outcomes. Demographics and established predictors of wellbeing were statistically controlled, and other aspects of behaviour related to COVID-19, such as wearing masks and fear of infection, were also included in the analyses. Data were collected using online survey technology. The following hypotheses were tested:

**Hypothesis** **1.***It was predicted that COVID-19-related information overload would have a negative effect and be positively associated with negative wellbeing outcomes and negatively associated with positive wellbeing outcomes*.

**Hypothesis** **2.***It was predicted that the established wellbeing predictors would be associated with wellbeing outcomes. PSYCAP and POSCOP and positive wellbeing should be associated, and GENIO and NEGCOP associated with negative wellbeing*.

**Hypothesis** **3.***The above hypotheses are related to specific direction effects based on the previous literature. The final hypothesis, for which there was no prior relevant literature, examined whether associations between COVID-19-related information overload and wellbeing were still significant when the established predictors of wellbeing were co-varied*.

## 2. Materials and Methods

### 2.1. The Survey

The survey ran from 10 February to 18 February 2020. Participants were given information about the study and an informed consent form on the first two pages of the online survey. If participants signed the consent forms, they were asked to answer the questionnaire. They were free to withdraw from the survey at any point. They were also informed that they had the right to refuse to answer any questions that made them feel uncomfortable. This study was reviewed and approved by the Health Science Centre Research Ethics Committee at Shenzhen University. 

### 2.2. Sample

A sample size calculation suggested that a sample of 1000 would be appropriate to detect effects of COVID-19-related information overload after adjustment for multiple covariates. A “snowball” sampling method was used in this study, focusing on cities or areas subjected to community containment measures during the initial phase of COVID-19. An anonymous volunteer organisation assisted in recruiting participants in residential areas. A total of 1349 participants completed the survey, and the disease status of the sample is shown in the Appendix A. A total of 1240 members of the general public with no known infection were analysed here (infected, ill and healthcare workers removed).

### 2.3. Demographics

The sample was 55.6% female and 97.9% Chinese speakers, and 49.4% of them were single. The age distribution was: 17–25 years: 26%; 26–30 years: 24.3%; 31–40 years: 23.9%; 41–50 years: 16.2%; 51 years+: 9.6%. The most frequent occupations were: 21.5% students; 19.5% teachers; 25.9% office workers; 10.8% managers, plus a few in a wide range of jobs. Frequencies in different cities/provinces are shown in the Appendix A. Frequencies in locations with different risks are shown in the Appendix A.

### 2.4. Measures

The assessment used in this study was based on the PIOS and WPQ, and the original English text in these scales was translated into Chinese. The translators were Chinese academics who had extensive postgraduate experience in the UK and previously written theses in English. Translation focused on getting the meaning of English to the equivalent meaning of Chinese and also focused on transferring cultural equivalence. A few nouns in the information overload survey were adapted to fit the local internet usage habits. For example, “WeChat” and “Weibo” (the name of the local social media platform) were used instead of “Facebook” or “Myspace”. The Chinese and English versions of the questionnaire used here are shown in the Appendix A.

#### 2.4.1. COVID-19 Behaviours: Information Overload, Time Spent Getting Information, Attention to COVID-19, Masks and Hygiene 

The first set of questions (COVID-TIME) measured the amount of time each day spent on getting information about COVID-19 from social media and news apps. For example, “In the last two weeks, what was the average hours per day that you spent on getting COVID19 information, through the following option? WhatsApp, Facebook, Messages”. The second set of questions measured information overload due to COVID-19 (COVID-IO). These were based on the questions in the PIOS, but the context changed to reflect the timing of COVID-19. For example, “In the last two weeks, how often have you received more information, updates, and case stories about COVID-19 than you can handle?” (Responses were on a scale from Never to Very often). Factor analyses were conducted to determine whether the COVID-TIME and COVID-IO measures consisted of one or more scales. Three individual questions asked about (1) the amount of attention paid to COVID-19 (COVID-ATT): “In the last two weeks, how often have you felt that you spend too much time on paying attention on COVID19?” 0—Never; 1—Almost never; 2—Sometimes; 3—Fairly often; 4—Very often” (2) Panic due to COVID-19 (PANIC): “Overall, to what extent you feel panic due to the COVID19 outbreaks?” Not at all, 1, 2, 3, 4, 5, 6, 7, 8, 9, 10, Very much so, and issues related to face masks and hand sanitisers (MASKS): “In the last two weeks, how often have you received the updated information about masks and disinfection supplies restocking?”

0—Never; 1—Almost never; 2—Sometimes; 3—Fairly often; 4—Very often. 

#### 2.4.2. Predictors of Wellbeing

(a)General information overload (GENIO)

These questions were from the PIOS and reflected generic information overload from media or other sources. For example, a media-related question was: “In the last two weeks, how often have you felt pressured to respond to messages, text or e-mail, quickly?” A question relating to overload from an environmental source was: “In the last two weeks, how often have you felt that the environment surrounding you is too noisy?”

It was predicted that this measure would be associated with negative wellbeing.

(b)Negative Coping (NEGCOP)

This measure was from the Wellbeing Process Questionnaire, and it was predicted that it would be associated with negative wellbeing.

(c)Positive Coping (POSCOP)

This measure was from the Wellbeing Process Questionnaire, and it was predicted that it would be associated with positive wellbeing.

(d)Psychological Capital (PSYCAP)

This measure was from the Wellbeing Process Questionnaire, and it was predicted that it would be associated with positive wellbeing.

#### 2.4.3. Wellbeing Outcomes

Items from the WPQ were used to measure positive wellbeing (POSWB: good mental health; life satisfaction and positive affect; Cronbach alpha = 0.74) and negative wellbeing (NEGWB: stress, negative affect, anxiety and depression; Cronbach alpha = 0.88).

### 2.5. Analysis Strategy

Factor analyses were carried out to confirm that the COVID-19 measures used in the subsequent regressions were appropriate. Correlations between the predictor variables and outcome measures were then examined. Separate regressions using the “ENTER” method were carried out on the total positive and negative outcome scores. The “ENTER” method was used as previous research has shown that the predictor variables have independent effects. The predictors consisted of demographic variables, established predictors of wellbeing and the COVID-19-specific measures. All analyses were carried out using the IBM SPSS package, version 25.2.

## 3. Results

The data from the study are available for use by other researchers (see link at the end of the paper).

### 3.1. Missing Data

There were very little missing data, and none of the missing values were replaced. Examination of outliers using stem-and-leaf plots from the SPSS “Explore” analysis suggested that these were not a major issue (72 outliers from 13,640 responses). SPSS considers any data value to be an outlier if it lies outside of the following ranges: 3rd quartile+ 1.5×interquartile range; 1st quartile−1.5×interquartile range.

### 3.2. Factor Analyses of COVID-TIME and COVID-IO Items

The usual method of combining items measuring time spent on various activities is to add them up. Similarly, total information overload is often seen as the sum of all the items on the scale. To confirm these general assumptions, separate factor analyses were carried out on the COVID-TIME and COVID-IO items. The factor analyses used a principal components method with varimax rotation. The output from these analyses is shown in Table 1 and Table 2. The Kaiser–Meyer–Olkin (KMO) values for both analyses were above 0.8, showing that the data were appropriate for factor analyses. Both the COVID-TIME analysis and the COVID-IO analyses led to single-factor solutions, with the factors accounting for 54% of the variance. Full output from the factor analyses is shown in the Appendix A. Total scores for these scales were, therefore, used in the regressions (COVID-TIME Cronbach alpha: 0.89; COVID-IO Cronbach alpha: 0.88). 

### 3.3. Correlations between Variables

#### 3.3.1. COVID-19 Variables and Wellbeing Outcomes

The correlations between the two wellbeing outcomes and the COVID-19 variables are shown in Table 3. This showed significant associations between all variables but with considerable variation in the size of the effects, and with the larger correlations being with negative wellbeing. These results confirm the prediction that the COVID-19-specific information overload measures are associated with the wellbeing outcomes.

#### 3.3.2. Established Predictors and Wellbeing Outcomes

The correlations between the established predictors and the wellbeing outcomes are shown in Table 4. These results show the usual pattern of associations between the predictors and wellbeing outcomes, apart from the significant positive correlation between positive coping and wellbeing. These results confirm the prediction made in Hypothesis 2.

### 3.4. COVID-19 Predictors and Established Wellbeing Predictors

The correlations between the established wellbeing predictors and the COVID-19 variables are shown in Table 5. There were significant correlations, especially for GENIO and PSYCAP.

In the next step of the analyses, testing Hypothesis 3, regressions were carried out to examine whether the associations between the COVID-19-specific variables and the wellbeing outcomes remained significant when the established predictors of wellbeing were controlled.

#### 3.4.1. Positive Wellbeing and COVID-19 Variables Controlling for Established Predictors

The first regression used POSWB as the dependent variable, and the output is shown in Table 6. The R^2^ value was 0.45 and F 12,1227 = 83.8 *p* < 0.001. All VIF values were less than 3, and the tolerance values greater than 0.2. The established predictors of positive wellbeing (PSYCAP; GENIO; POSCOP) had their usual significant effects, with higher PSYCAP and POSCOP being associated with greater positive wellbeing and higher GENIO being associated with lower positive wellbeing. NEGCOP showed an unusual effect, being positively associated with positive wellbeing. Possible reasons for this are covered in the discussion. With regard to the COVID-19-related predictors, spending time getting information about COVID-19 (COVID-TIME) was associated with positive wellbeing. In contrast, COVID-IO and PANIC were associated with reduced positive wellbeing.

#### 3.4.2. Negative Wellbeing and COVID-19 Variables Controlling for Established Predictors

The second regression used negative wellbeing (NEGWB) as the dependent variable, and the results are shown in Table 7. The R^2^ value was 0.55 and F 12,1227 = 125.7 *p* < 0.001. Again, the established predictors of negative wellbeing outcomes (PSYCAP, GENIO, POSCOP and NEGCOP) had their usual significant effects. Higher GENIO and NEGCOP scores were associated with greater negative wellbeing and higher PSYCAP and POSCOP with lower negative wellbeing. With regard to the COVID-19 predictors, PANIC, COVID-ATT and COVID-IO were positively associated with negative wellbeing. Age was negatively associated with negative wellbeing (older participants reported less stress and negative affect).

### 3.5. Analyses of Covariance Examining COVID-19 Predictors and Established Wellbeing Predictors

Another method of examining Hypothesis 3 involved analyses of covariance with the wellbeing measures as the dependent variables, individual COVID-19 variables as the independent variable, and the established predictors and other COVID-19 variables as covariates. The COVID-19 variables were split into tertiles (low, medium and high scores) to give an indication of dose–response, which is a better indicator of causality. The complete set of analyses is in the Appendix A. Of the COVID-19 variables, only COVID-IO showed significant effects (positive wellbeing: F2, 1229 = 8.94 *p* < 0.001 partial eta squared = 0.014; negative wellbeing: F2, 1229 = 59.93 *p* < 0.01 partial eta squared = 0.088) and the means (shown in Table 8 show linear dose–response. 

## 4. Discussion 

The present study examined wellbeing during the initial phase of the COVID-19 pandemic in China after community containment measures were imposed. At the time the study was conducted, there were no published studies on COVID-19 and information overload. A PubMed literature search now shows that since that time, 60 studies have been conducted on this topic. Some of these published studies have looked at other aspects of information overload and not its association with wellbeing. For example, some studies have examined how other measures of information overload (e.g., the cancer information overload scale) can be adapted for the COVID-19 context [43]. Other research has examined the incidence of COVID-19-related information overload [44]. Research has also tried to identify risk factors for the occurrence of information overload [45,46] and the impact it has on mental health [47]. These studies on mental health are the nearest in content to the present one. However, the published research has not controlled for the established predictors of mental health, and the present study makes a novel contribution in that it considers both positive and negative wellbeing outcomes and adjusts for possible confounding factors. Another literature search, this time using Psycinfo, revealed that there were no published studies on COVID-19, information overload and wellbeing (or mental health, or stress). A search of PubMed revealed three articles on these topics, but most of these were related to the content of the information and are discussed in a later section.

A secondary aim of the present study was to determine whether predictions from the wellbeing process model could be replicated in a Chinese sample. The majority of the research using the WPQ has been conducted in the UK. Two studies in other countries, one in Kuwait [48] and one in Kazakhstan [49], have replicated the results from the UK. The results from the present study showed that the established predictors of either positive or negative wellbeing were significant in this study, confirming previous findings [34,35,36,37,38] and showing that the model generalised to a Chinese sample. The one predictor that showed an unusual profile of associations was negative coping, which was positively associated with both negative outcomes (the usual finding) and positive outcomes. There are a number of possible reasons for this result. First, coping may be a more global concept for the Chinese, and this was supported by a small positive correlation between positive and negative coping (usually, there is a negative correlation). Secondly, the concept of negative coping may have been lost in translation. Further research is needed to address this issue, which is not the only time that differences in the effects of psychosocial variables have been observed with Chinese samples. In an unpublished study, we found that social support has a very different meaning to the Chinese and in that country reflects a reliance on social services. 

One aim of the present research was to examine whether specific COVID-19 behaviours also had significant associations with wellbeing. Spending time getting information about COVID-19 was associated with greater positive wellbeing. COVID-IO and PANIC were associated with reduced (more negative) wellbeing. These results suggest that in order to maintain their wellbeing, individuals should avoid information overload and reduce the time spent on, or take a break from, the news and social media information related to COVID-19. Actively searching for information about COVID-19, rather than being exposed to it from sources outside of one’s control, appears to be good for wellbeing. These findings support recent research [50], which examined the WHO view that misinformation through social media is a major threat to an appropriate COVID-19 response. 

Correct information about COVID-19 is central in dealing with the pandemic, and the present study shows that spending time obtaining appropriate information on the topic is beneficial for the person’s wellbeing. Other research [51] confirms that it is essential to use methods of disseminating relevant information about COVID-19 without increasing the risk of information overload. The content of the information should also be considered, and “Fake News” is also very prevalent, and it is important to reduce this to maintain a balanced perception of the pandemic. It was not the aim of the present research to investigate the content of the information. However, three recent articles have done this, and these findings are relevant here. The first paper [52] describes the “infodemic” in the COVID-19 pandemic in terms of false news, conspiracy theories, magical cues and racist comments. This information has been increasing at an alarming rate and has the potential to increase stress and anxiety and even lead to loss of life. The second study [44] demonstrated that COVID-19 information is often conflicting and can lead to confusion, which in turn can have unfavourable effects on the prevention and management of the pandemic. The last study [53] shows that one should not consider information on its own but must consider it in combination with isolation, stress and other risk factors, such as physical inactivity, which create vicious circles accelerated by COVID-19. 

Other research has focused on loneliness during COVID-19, and the negative effects of social isolation can potentially be reduced by social media and the internet. However, this needs to be done in a controlled way, as loneliness can also increase problematic internet use [54], which can lead to increased distress [55]. Indeed, as described above, there are a number of risk factors for reduced metal health during the pandemic, and it is important to include these other risks when investigating the possible negative effects of COVID-19. The present study showed that feelings of panic (PANIC) and COVID-IO had independent effects that led to reduced wellbeing. Future research must include other known risks such as job insecurity and risk of personal infection and infection of family/friends.

The present results have implications for the dissemination of information during pandemics. This information should have a clear message supported by science, and be presented in a concise, easy to understand message. Misinformation, lack of clarity and excessive information should be avoided. However, the present study has a number of limitations. The first limitation is that the study was only conducted in one country, and there is a need to replicate the present results using samples in countries where different approaches to the prevention and management of COVID-19 are used (e.g., less emphasis on lockdown and greater track-and-trace approaches). The present study had a cross-sectional design, and further longitudinal research is required as this will provide a better indication of causality. Further, the sampling method meant that it is highly unlikely that the present sample is representative of the entire Chinese population (many of whom do not use social media), and there is no easily accessible information about the characteristics of regular social media users in China to allow analysis of a representative sub-sample.

## 5. Conclusions

In conclusion, the present survey identified information overload about COVID-19 as a predictor of reduced wellbeing during isolation. This result was significant when established predictors of wellbeing were statistically controlled. The survey used short measuring instruments that can easily be used in future risk assessments. The results have implications for information campaigns and COVID-19-related communications. The major limitation of the study was that the survey was cross-sectional, which makes it difficult to assess causality. Another limitation was that the survey was the sampling method and the fact that it was conducted in only one country, and it is important to determine whether effects generalise to other contexts. Further information on other factors influencing wellbeing (e.g., job insecurity, perceived risk of infection) also need to be measured in future studies.

## Figures and Tables

**Figure 1 behavsci-11-00062-f001:**
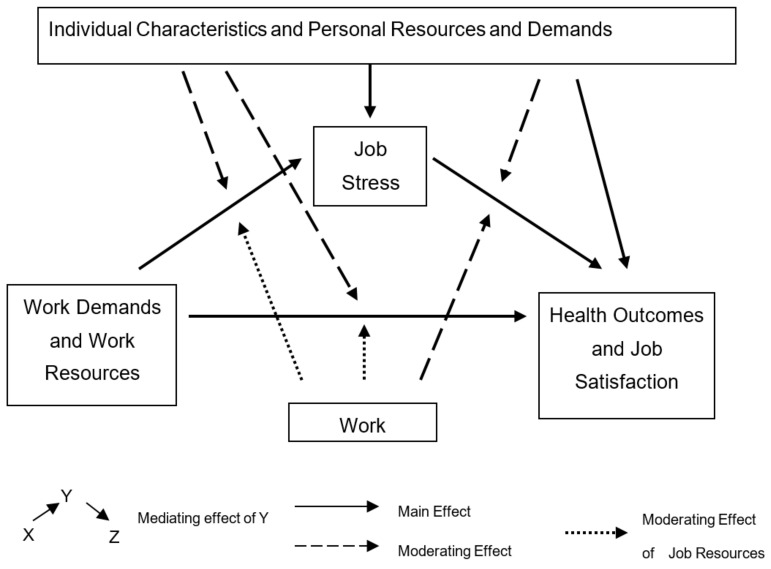
The Demands-Resources-Individual Effects Model.

**Figure 2 behavsci-11-00062-f002:**
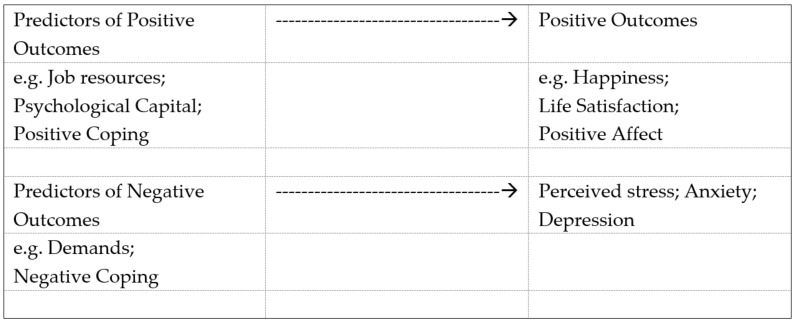
The wellbeing process model.

**Table 1 behavsci-11-00062-t001:** Factor analyses of the COVID-TIME variables: Component matrix.

Time_website	0.787
Time_TV	0.780
Time_otherSNS	0.771
Time_BCnewspaper	0.770
Time_OnlineNews	0.765
Time_OnlineVideo	0.737
Time_Other	0.711
Time_QQ	0.709
Time_wechat	0.677
Time_weibo	0.543

**Table 2 behavsci-11-00062-t002:** Factor analyses of the COVID-IO variables: Component matrix.

Q4_pressured_to_manage_at_same_time	0.820
Q6_instant_messages	0.819
Q3_spend_too_much_time	0.759
Q2_received_too_much_info	0.735
Q8_lower_sensitivity	0.735
Q5_online_social_network	0.727
Q9_additional_demands	0.676
Q10_escape	0.672
Q7_less_time_for_leisure	0.656
Q1_overwhelmed	0.633

Key: Wording of the COVID-IO questions. Q1: In the last two weeks, how often have you felt overwhelmed with the COVID19 updates you received? Q2: In the last two weeks, how often have you received more information, updates, cases stories about COVID19 than you can handle? Q3: In the last two weeks, how often have you felt that you spend too much time paying attention to COVID19? Q4: In the last two weeks, how often have you felt pressured to manage several information and communication inputs about COVID19 at the same time? Q5: In the last two weeks, how often have you felt that you received too many COVID19 updates (e.g., news, applications pop-up, event notifications, personal messages, and status updates) to deal with? Q6: In the last two weeks, how often have you felt that you have received more instant messages about COVID19 that you can handle? Q7: In the last two weeks, how often have you felt that you are focusing on COVID19 leaves you too little time for recreational activities? Q8: In the last two weeks, how often have you felt that your focusing on COVID19 information makes you less sensitive to the needs of others? Q9: In the last two weeks, how often have you felt that the extra demands on you due to COVID19 exceed your capacity to deal with them? (e.g., looking for masks and disinfection supplies, doing disinfection cleaning, extra work tasks, overtime work). Q10: In the last two weeks, how often have you felt that you wanted to escape from the network and media to reduce your attention on COVID19?

**Table 3 behavsci-11-00062-t003:** Correlations between predictor variables and wellbeing outcomes (significant associations).

Variable	Positive Wellbeing	Negative Wellbeing
COVID-IO	−0.295 **	0.653 **
MASKS	−0.198 **	0.438 **
COVID-TIME	0.092 *	0.131 **
COVID-ATT	0.015 ns	0.118 **
PANIC-COVID	−0.355 **	0.533 **

** *p* < 0.001, * *p* < 0.005.

**Table 4 behavsci-11-00062-t004:** Correlations between predictor variables and wellbeing outcomes (usual pattern of associations).

Variable	Positive Wellbeing	Negative Wellbeing
PSYCAP	0.567 **	−0.391 **
POSCOP	0.439 **	−0.090 **
NEGCOP	0.114 **	0.228 **
GENIO	−0.250 **	0.537 **

** *p* < 0.001.

**Table 5 behavsci-11-00062-t005:** Correlations between COVID-19 variables and established predictors of wellbeing outcomes.

Variable	GENIO	PSYCAP	POSCOP	NEGCOP
COVID_TIME	0.229 **	0.121 **	0.113 **	0.029
COVID_ATT	0.056 *	0.087 **	0.095 **	0.006
PANIC	0.345 **	−0.348 **	−0.040	0.050
COVID_IO	0.717 **	−0.263 **	−0.034	0.141 **
MASKS	0.530 **	−0.207 **	0.041	0.081 *

** *p* < 0.001, * *p* < 0.05.

**Table 6 behavsci-11-00062-t006:** Predictors of positive wellbeing.

Model	Unstandardised Coefficients		Standardised Coefficients	t	Sig.
	B	Std. Error	Beta		
(Constant)	9.137	1.080		8.462	0.000
GENDER	0.187	0.230	0.018	0.814	0.416
AGE	0.076	0.129	0.019	0.588	0.557
MARITAL STATUS	−0.140	0.169	−0.027	−0.827	0.408
PSYCAP	0.392	0.026	0.381	14.854	0.000
POSCOP	0.736	0.062	0.276	11.818	0.000
NEGCOP	0.176	0.045	0.085	3.874	0.000
GENIO	−0.115	0.054	−0.067	−2.134	0.033
COVID_IO	−0.060	0.022	−0.100	−2.787	0.005
MASKS	0.025	0.048	0.014	0.511	0.610
COVID_TIME	0.031	0.010	0.071	3.091	0.002
COVID_ATT	−0.083	0.167	−0.011	−0.494	0.621
PANIC	−0.384	0.063	−0.157	−6.061	0.000

**Table 7 behavsci-11-00062-t007:** Predictors of negative wellbeing.

Model	Unstandardised Coefficients		Standardised Coefficients	t	Sig.
	B	Std. Error	Beta		
(Constant)	5.643	1.654		3.413	0.001
GENDER	−0.144	0.352	−0.008	−0.408	0.683
AGE	−0.556	0.197	−0.083	−2.820	0.005
MARITAL STATUS	0.096	0.259	0.011	0.372	0.710
PSYCAP	−0.271	0.040	−0.155	−6.699	0.000
POSCOP	−0.213	0.095	−0.047	−2.230	0.026
NEGCOP	0.530	0.069	0.152	7.623	0.000
GENIO	0.383	0.083	0.131	4.627	0.000
COVID_IO	0.383	0.033	0.373	11.535	0.000
MASKS	−0.004	0.074	−0.001	−0.056	0.956
COVID_TIME	−0.004	0.016	−0.005	−0.238	0.812
COVID_ATT	0.521	0.256	0.042	2.031	0.042
PANIC_ COVID	0.956	0.097	0.230	9.848	0.000

**Table 8 behavsci-11-00062-t008:** COVID-IO and wellbeing (scores are the means, s.d.s in parentheses).

	Positive Wellbeing	Negative Wellbeing
Low COVID-IO	22.03 (4.83)	13.20 (6.71)
Medium COVID-IO	19.57 (5.00)	18.64 (7.37)
High COVID-IO	18.12 (5.23)	26.69 (6.97)

## Data Availability

Data have been deposited in ResearchGate. https://www.researchgate.net/publication/344172804_Data_for_information_overload_and_covid (accessed on 26 April 2021).

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
