# Peer review of "Information Overload, Wellbeing and COVID-19: A Survey in China"

_behavsci, 2021, doi:10.3390/bs11050062_

Round 1

Reviewer 1 Report

Dear authors,

Contributions to our understanding of psychological well-being during the COVID-19 pandemic are appreciated. The concept of information overload is a valuable construct in this regard. The nature of information and how it is processed by individuals is highly critical, due to the potential for problematic internet-related use that can lead to increased stress. Thus, there are several studies published in the last year which discuss the nature of information seeking, social media sources, and mobile device use for access to information on COVID-19, and the resulting (usually negative) impacts on psychological well being. I offer suggestions for revisions to the manuscript below.

  1. Theoretical framework: You state that the "wellbeing process model" (lines 86-87) serves as your theoretical framework. The history of the model is briefly mentioned. However, there should be far greater detail provided on this model. A visualization of the model itself would be immensely helpful to readers. Furthermore, each of the factors of the model should be discussed. This should, in turn, lead to some research questions and hypotheses (below).
  2. Hypothesis development: While it is stated in the manuscript that certain expected relationships were observed (lines 195-198, 205-206, 219-221). However, there is no section for the development of testable hypotheses. Nor are the results and discussion sections framed around any clear research goals.
  3. Sample and data collection: The demographics (lines 132-137) are provided. However, since a snowball approach to data collection was adopted, how can we know if the data are representative of Chinese respondents overall? There was "very little missing data" (line 181). However, did you evaluate the data for outliers?
  4. Instrumentation: There are several issues with the presentation of your measures (page 4). 1) More detailed descriptions of many variables should be provided, including sample items (such as for MASK, COVID-ATT, PANIC, PSYCAP, POSCOP, NEGCOP, GENIO). 2) These measures are not mentioned in the introduction or clearly linked to the theoretical framework (which itself is lacking in clarity and detail). 3) The validity, reliability, internal consistency, etc. for the scales should be mentioned in the Methods section.  4) The process of translating items to Chinese is lacking in sufficient evidence that the resulting scale could be validated. What are the qualifications of the translators? What are the steps used to establish cultural or linguistic equivalence?
  5. Factor Analysis: There is no graphic representation of the results, which would be of interest to readers, particularly if your PIOS and WPQ translated scales could constitute a contribution to the literature regarding psychometrics. Were the assumptions necessary for FA met? Was multicollinearity ruled out? Where is a correlation matrix of the variables included in your analyses?
  6. Regression: It is assumed that the variable selection method of "enter" was used (including all variables). Were other variable selection methods considered? Given your proposed framework, why weren't alternative models compared? The representation of these results seems a bit too basic, and the interpretation can be difficult if there is only a single model provided. There are no surprising results, which begs the question of what your specific contribution could be. There likely is one, but it is not clearly argued. In Table 1, why is the beta coefficient for NEGCOP positive when predicting POSWB? The text states that this was a negative association (lines 197-198). R2 is not stated.
  7. Interpretation: There are too few citations in the discussion and conclusion section, and there is no clear link of the interpretation of the findings to your theoretical framework. Many of the results are expected and have already been established in the literature. The main point of interest is the role of information and information overload in terms of psychological well-being. Discussion of this point requires some discussion of the many findings in the literature which find negative relationships between the time spent reading information on COVID-19 and one's psychological well-being. In fact, stress is often either a predictor of time (often problematic amounts of time) spent reading about COVID-19 - in that this kind of information searching behavior is a mechanism for dealing with the stress of lockdown or uncertainty - or that time spent on reading about COVID-19 is a predictor of stress - in that reading more troublesome news results in a vicious cycle leading to increases in stress and, arguably, information overload. Overall, the discussion should draw from some opposing or contradictory findings in the past 6 months or so, and should be more nuanced in terms of factors such as "fake news," "misinformation," and related negative sources of information.
  8. References: The formatting is not consistent and needs to be checked.
  9. Self-citation: 8 self-citations seems excessive for one of the co-authors.
  10. Plagiarism: some self-plagiarism should also be fixed/paraphrased (lines 11-13, 46-49, 69-83, 86-98)

Author Response

We thank the reviewer for the constructive comments. We have addressed the specific points as follows and a version of the paper with changes marked is included in the submission.

  1. More information about the wellbeing process model is now provided, and the individual predictors discussed. The hypotheses tested are now made explicit.
  2. Each section of the paper now covers the information in a structure related to the hypotheses.
  3. There is now further coverage of sampling issues in the discussion. Information on outliers is now provided.
  4. Examples of the items are now given in the text (the complete survey – both Chinese and English versions – is in the supplementary material.
  5. The results from the factor analyses are now shown in the text, with appropriate statistics covering the appropriateness of factor analysis and multicollinearity.
  6. The regressions have now been re-done to include demographic variables (as suggested by another reviewer). The “ENTER” method is used as the model suggests direct relationships between the predictors and outcomes. The R2 values and other relevant statistics have been added. The unusual result for NEGCOP is covered in the discussion. Similarly, other research on COVD-19 related information overload is now discussed, and it is shown that this is a uniques studying demonstrating that the in formation overload reduces wellbeing and this is still observed when established predictors of wellbeing are controlled.
  7. The discussion now put the present study in the context of the literature which has been published after the reported study was conducted.
  8. The formatting of the references has been checked.
  9. Self-citation is a major issue when authors cite their own work rather than relevant research by others. In the case of the DRIVE model and wellbeing process model there is not yet a literature from other groups.
  10. The duplicate sentences have now been removed.

Reviewer 2 Report

This is an interesting paper. The sample is acceptable and the data collected supports the creation of the variables. I liked the idea of collection data on general information overload (GENIO) and data on time searching COVID19 information. It made a stronger argument that collection of data on only one of the components. 

I have a number of questions and comments that may improve the quality of the manuscript.

  1. In the introduction I think that "the city most affecting..." should be changed by the "city most affected"
  2. In the introduction, the concept of community containement measures was not clear to me. Can you define precisely what is community containment measures? pay attention that I am not mentioning individual precautionary behaviors, but the community level measures and not the individual level measures
  3. The dependent variables of the study were positive and negative well being. I wonder why the socio demographic variables were not included in the model? We know that there is a link between well being and education, marital status and age. Don't you think that you need to control for these variables.?
  4. In table 1, the results for the association of two independent variables and positive well being are not clear to me. Positive coping and negative coping both have a positive and statistically significant effect on positive well being. does it makes sense that negative coping has a positive effect on positive well being? I was thinking that does not make much sense. but I am curious to hear the author/s opinion.
  5. On related issue, to my 4th point, on Table 2, positive coping is negative related to negative well being. And negative coping is positively related to negative well being. Can you explain the disparity between my 4th point and my 5th point?
  6. In the discussion, the authors indicate the "importance of receiving accurate information" but  I am not sure this is something you can infer from the findings

Author Response

We thank the reviewer for the constructive comments. We have addressed the points made by the reviewer in the following way and a version of the paper with changes marked is included in the submission.

  1. The first paragraph has been deleted at the request of another reviewer, which now makes this point redundant.
  2. The first paragraph has been deleted at the request of another reviewer, which now makes this point redundant.
  3. Demographic variables have now been included in the regression.
  4. The issue of the unusual effect of negative coping is covered in the discussion.
  5. The issue of the unusual effect of negative coping is covered in the discussion.
  6. The importance of considering the content of the information, not just the amount, is now discussed as a distinct topic.

Reviewer 3 Report

COVID-19 related studies are conducted extensively in recent months. The present study is in line with the general interest in the effects of the pandemic. It is the correlational, cross-sectional study which in fact confirms already known relationships, but in the COVID-19 related context. The way the manuscript is presented requires some improvements, which I list below:  

  1. The first para (lines 36-49) can be removed as it adds little new to knowledge on COVID-19.
  2. Some parts of the text are misplaced or duplicated - e.g. lines 78-85 should be moved to Measures section, lines 257-261 repeat information given in lines 243-252.
  3. Lines 118-119 - it is not clear what the Authors actually mean and how such circumstances affect the study.
  4. Analysis strategy should be described in more details - what kind of regression was used and why, in which way the regression equation was constructed
  5. Some phrases are not clear or confusing - is line 221 related to theoretical or to regression model? In lines 222-225 second sentence contradicts the first one; information in line 235-236 is irrelevant as social isolation was not studied
  6. Line 243-246 - the study did not look at the content of information thus the comment is not based on the results
  7. Line 248  - China used similar measures to  many other countries. What countries of different approach are ment? How this different approach might affect information overload or wellbeing?

  8. More information is required on the theoretical model of the study and hypotheses - part 1.1. needs elaboration and improvement as it is crucial for the study.

Author Response

We thank the reviewer for the constructive comments. We have addressed the points made by the reviewer in the following way and a version of the paper with changes marked is included in the submission.

  1. The first paragraph has been deleted.
  2. The misplaced lines have been moved and the duplicate lines deleted.
  3. These lines have now been deleted.
  4. More information is now provided about the analysis strategy.
  5. Line 221 refers to the wellbeing process model and this has now been made explicit.
  6. The importance of considering the content of the information, not just the amount, is now discussed as a distinct topic.
  7. There are different strategies used to prevent and manage COVID-19. In some countries, complete lockdown is the main strategy. Others rely more on social distancing and track-and-trace. Different messages are associated with these strategies and hence the need to examine such issues across different countries.
  8. More information about the underlying model has now been given.

Round 2

Reviewer 1 Report

We thank the reviewer for the constructive comments. We have addressed the specific points as follows and a version of the paper with changes marked is included in the submission.

Reviewer follow-up: I appreciate the time and effort in making changes in response to my concerns. I have reorganized your comments beneath my original review for easier reference. Some points have been addressed, but others still remain unresolved. Please refer to my follow-up comments below:

  1. Theoretical framework: You state that the "wellbeing process model" (lines 86-87) serves as your theoretical framework. The history of the model is briefly mentioned. However, there should be far greater detail provided on this model. A visualization of the model itself would be immensely helpful to readers. Furthermore, each of the factors of the model should be discussed. This should, in turn, lead to some research questions and hypotheses (below).

Authors’ response: More information about the wellbeing process model is now provided, and the individual predictors discussed. The hypotheses tested are now made explicit.

Reviewer follow-up:

a) I cannot find where additional information on the wellbeing process model are provided in the introduction.

b) No visualization was provided, which would allow the reader to better understand the relationships among factors in the model (wellbeing process model).

c) The predictor variables are mentioned, but are not defined or linked to any literature in the introduction.

d) There is no clear link provided between the wellbeing process model and your selected predictor variables. The relationship of your questionnaire items (constructs and variables) to the theoretical model should be explicitly described.

e) The hypotheses are not clearly presented. By that I mean that they should be displayed on their own line, and in conjunction with research questions which are evidently linked to your literature review.

f) Overall, the literature review is shorter, rather than longer, and not enough elaboration of the wellbeing process model or your selected predictors is provided. For example, the new content (lines 105-178) is too short and lacking in citations in regards to your predictor variables.

  1. Hypothesis development: While it is stated in the manuscript that certain expected relationships were observed (lines 195-198, 205-206, 219-221). However, there is no section for the development of testable hypotheses. Nor are the results and discussion sections framed around any clear research goals.

Authors’ response: Each section of the paper now covers the information in a structure related to the hypotheses.

Reviewer follow-up:

a) The methods section should be more clearly labelled to indicate which construct is being measured, rather than by including the name of the scales. This is what I meant by consistent reference to the hypothesized predictors.

b) There is still no use of subheadings in the Results section (which is organized according to the type of analysis used, rather than the research questions/hypotheses) or the Discussion section (which includes a discussion of the research gap and how you addressed it [paragraph 1], replication of prior findings with a Chinese population [paragraph 2], the role of specific predictors [paragraphs 3 – 5] and the implications [paragraph 6] and conclusions/limitations [paragraph 7]) to indicate that the presentation is based on your hypotheses, consistently discussed in the order they are presented.

  1. Sample and data collection: The demographics (lines 132-137) are provided. However, since a snowball approach to data collection was adopted, how can we know if the data are representative of Chinese respondents overall? There was "very little missing data" (line 181). However, did you evaluate the data for outliers?

Authors’ response: There is now further coverage of sampling issues in the discussion. Information on outliers is now provided.

Reviewer follow-up:

a) It is still not clear (lines 304 – 308) what the threshold for identifying outliers was or if the outlier data were deleted.

b) I notice that you mentioned that “the sampling method meant that it is highly unlikely that the present sample is representative of the Chinese population” (lines 665 – 666). Shouldn’t that be a major cause for concern? Is it possible to reanalyse your data set by selecting cases based on a sampling framework that would assist in achieving representativeness?

  1. Instrumentation: There are several issues with the presentation of your measures (page 4). 1) More detailed descriptions of many variables should be provided, including sample items (such as for MASK, COVID-ATT, PANIC, PSYCAP, POSCOP, NEGCOP, GENIO). 2) These measures are not mentioned in the introduction or clearly linked to the theoretical framework (which itself is lacking in clarity and detail). 3) The validity, reliability, internal consistency, etc. for the scales should be mentioned in the Methods section. 4) The process of translating items to Chinese is lacking in sufficient evidence that the resulting scale could be validated. What are the qualifications of the translators? What are the steps used to establish cultural or linguistic equivalence?

Authors’ response: Examples of the items are now given in the text (the complete survey – both Chinese and English versions – is in the supplementary material.

Reviewer follow-up:

a) As mentioned above, the variables are not clearly defined in the introduction or linked to empirical studies to support their inclusion.

b) While some sample items were added for COVID-19 behaviors, there are still no validity, reliability, or internal consistency values or explanations stated for any variables (apart from the Wellbeing Process Questionnaire and PIOS).

c) The full list of questions for the COVID-IO construct is provided on lines 334-352. I suggest just included a couple of examples in the Materials and Method section instead.

d) Factor analysis seems to support the unidimensionality of the constructs COVID-TIME and COVID-IO. However, there is not really enough detail provided. I suggest checking the following paper by Slocum-Gori and Zumbo (2011): https://doi.org/10.1007/s11205-010-9682-8

e) The description of the translation process has been improved, but there is still no details on where and how items were revised and evaluated in terms of linguistic or cultural equivalence. The explanation of this procedure is important for readers, particularly since you are providing the scale as supplementary material. However, the Chinese version should be included as well, so that it could be used by other researchers studying related constructs with Chinese populations.

d) There are still no sample items for PIOS in the manuscript.

  1. Factor Analysis: There is no graphic representation of the results, which would be of interest to readers, particularly if your PIOS and WPQ translated scales could constitute a contribution to the literature regarding psychometrics. Were the assumptions necessary for FA met? Was multicollinearity ruled out? Where is a correlation matrix of the variables included in your analyses?

Authors’ response: The results from the factor analyses are now shown in the text, with appropriate statistics covering the appropriateness of factor analysis and multicollinearity.

Reviewer follow-up:

a) There is still no full correlation matrix. Table 3 only contains correlations between the predictor and outcome variables. A full matrix is preferred, particularly in establishing the rationale and meeting the assumptions for regression analysis.

b) No values for statistical significance are provided for the correlations.

c) As mentioned above, the discussion of FA and relevant statistics or assumption tests are not complete.

  1. Regression: It is assumed that the variable selection method of "enter" was used (including all variables). Were other variable selection methods considered? Given your proposed framework, why weren't alternative models compared? The representation of these results seems a bit too basic, and the interpretation can be difficult if there is only a single model provided. There are no surprising results, which begs the question of what your specific contribution could be. There likely is one, but it is not clearly argued. In Table 1, why is the beta coefficient for NEGCOP positive when predicting POSWB? The text states that this was a negative association (lines 197-198). R2 is not stated.

Authors’ response: The regressions have now been re-done to include demographic variables (as suggested by another reviewer). The “ENTER” method is used as the model suggests direct relationships between the predictors and outcomes. The R2 values and other relevant statistics have been added. The unusual result for NEGCOP is covered in the discussion. Similarly, other research on COVD-19 related information overload is now discussed, and it is shown that this is a uniques studying demonstrating that the in formation overload reduces wellbeing and this is still observed when established predictors of wellbeing are controlled.

Reviewer follow-up:

a) It is nice to see the VIF statistics mentioned.

b) I am confused by the mention of Change in F (line 395), as this seems to suggest that you compared the R2 between the models for positive wellbeing as a DV and negative wellbeing as a DV. This is hard to understand.

c) Certainly it would make more sense to compare two models (one with the control variables and one without). In this context, the change in F would be a meaningful result that could be interpreted.

d) It is not convincing to state that “information overload reduces wellbeing and this is still observed when established predictors of wellbeing are controlled.” I think that you might have a contribution here, since the standardized coefficient and significance of the predictive power of information overload is demonstrated in the overall model (for negative wellbeing, particularly). However, the wording can be more precise and the argumentation and explanation more clear. Further analyses, such as comparison of high, medium, and low information overload participants in terms of other factors (such as through ANOVA) could provide better insights into the relationships among variables.

e) There is no clear rationale for including the control variables (such as hypothesized effects on wellbeing), which should either be briefly discussed in the literature review or mentioned in the Methods or Results in terms of the regression analyses conducted.

  1. Interpretation: There are too few citations in the discussion and conclusion section, and there is no clear link of the interpretation of the findings to your theoretical framework. Many of the results are expected and have already been established in the literature. The main point of interest is the role of information and information overload in terms of psychological well-being. Discussion of this point requires some discussion of the many findings in the literature which find negative relationships between the time spent reading information on COVID-19 and one's psychological well-being. In fact, stress is often either a predictor of time (often problematic amounts of time) spent reading about COVID-19 - in that this kind of information searching behavior is a mechanism for dealing with the stress of lockdown or uncertainty - or that time spent on reading about COVID-19 is a predictor of stress - in that reading more troublesome news results in a vicious cycle leading to increases in stress and, arguably, information overload. Overall, the discussion should draw from some opposing or contradictory findings in the past 6 months or so, and should be more nuanced in terms of factors such as "fake news," "misinformation," and related negative sources of information.

Authors’ response: The discussion now put the present study in the context of the literature which has been published after the reported study was conducted.

Reviewer follow-up:

a) This aspect has been improved in the Discussion and Conclusions. It would be a good idea to include some of that relevant discussion in the introduction, to contextualize your study. While these sources may have been published after you concluded your study, they can still be retrospectively adapted to establish a more clear research gap and rationale.

b) I respectfully disagree with the claim that “Most of these published studies have looked at some other aspect of information overload and not its association with wellbeing” (lines 587-588) as I have published on this precise topic.

c) Again, a more nuanced approach (i.e., the nature of information which is being assessed for the construct of information overload) is strongly encouraged.

  1. References: The formatting is not consistent and needs to be checked.

Authors’ response: The formatting of the references has been checked.

Reviewer follow-up: Thank you for attending to this issue. Actually, the journal can assist in this, if accepted.

  1. Self-citation: 8 self-citations seems excessive for one of the co-authors.

Authors’ response: Self-citation is a major issue when authors cite their own work rather than relevant research by others. In the case of the DRIVE model and wellbeing process model there is not yet a literature from other groups.

Reviewer follow-up: Citations 30 through 35 might not be absolutely necessary, unless the DRIVE model can be more clearly shown and explained (which was not done). That would justify the inclusion of several of Dr. Smith’s seminal works in the field.

  1. Plagiarism: some self-plagiarism should also be fixed/paraphrased (lines 11-13, 46-49, 69-83, 86-98)

Authors’ response: The duplicate sentences have now been removed.

Reviewer follow-up: Thank you for attending to this issue.

Additional feedback

  1. Please use more sub-headings (particularly related to your key research questions).
  2. In addition to hypotheses, clear research questions in a separate sub-section at the end of the introduction would be helpful. This can help to provide a more clear framework for the writing of each section (consistently).
  3. The supplemental material is quite long and does not clearly divide the questions into sections which can be used in reference to the instruments mentioned in the Methods section. Clearer formatting of this file, if it is to be included with your submission, is required.

Author Response

Response to reviewer:

We thank the reviewer for the additional comments. Our reply is shown below.

Reviewer follow-up:

  1. I cannot find where additional information on the wellbeing process model are provided in the introduction.

This is now marked on page 3.

  1. No visualization was provided, which would allow the reader to better understand the relationships among factors in the model (wellbeing process model).

The DRIVE model and Wellbeing Process model are now shown on page 3.

  1. The predictor variables are mentioned, but are not defined or linked to any literature in the introduction.

The predictor variables are described on page 4.

  1. There is no clear link provided between the wellbeing process model and your selected predictor variables. The relationship of your questionnaire items (constructs and variables) to the theoretical model should be explicitly described.

The constructs and variables are now linked to the model (page 4).

  1. The hypotheses are not clearly presented. By that I mean that they should be displayed on their own line, and in conjunction with research questions which are evidently linked to your literature review.

The hypotheses are shown in section 1.4.

  1. Overall, the literature review is shorter, rather than longer, and not enough elaboration of the wellbeing process model or your selected predictors is provided. For example, the new content (lines 105-178) is too short and lacking in citations in regards to your predictor variables.

More detail and citations for the inclusion of the predictor variables in the model are now given (page 4).

Reviewer follow-up:

  1. The methods section should be more clearly labelled to indicate which construct is being measured, rather than by including the name of the scales. This is what I meant by consistent reference to the hypothesized predictors.

Better labelling is given in the method sections.

  1. b) There is still no use of subheadings in the Results section (which is organized according to the type of analysis used, rather than the research questions/hypotheses) or the Discussion section (which includes a discussion of the research gap and how you addressed it [paragraph 1], replication of prior findings with a Chinese population [paragraph 2], the role of specific predictors [paragraphs 3 – 5] and the implications [paragraph 6] and conclusions/limitations [paragraph 7]) to indicate that the presentation is based on your hypotheses, consistently discussed in the order they are presented.

More sub-headings are provided in all sections.

Reviewer follow-up:

  1. It is still not clear (lines 304 – 308) what the threshold for identifying outliers was or if the outlier data were deleted.

This information is now provided  - lines 271-4.

  1. I notice that you mentioned that “the sampling method meant that it is highly unlikely that the present sample is representative of the Chinese population” (lines 665 – 666). Shouldn’t that be a major cause for concern? Is it possible to reanalyse your data set by selecting cases based on a sampling framework that would assist in achieving representativeness?

This issue is covered in the discussion. The sampling method, through social media, means that the sample is not representative of the Chinese population as a whole. There is not relevant information to know the characteristics of the Chinese social media users to conduct an analysis on a representative sample.

 Reviewer follow-up:

  1. As mentioned above, the variables are not clearly defined in the introduction or linked to empirical studies to support their inclusion.

This is now done in the introduction (page 4).

  1. b) While some sample items were added for COVID-19 behaviors, there are still no validity, reliability, or internal consistency values or explanations stated for any variables (apart from the Wellbeing Process Questionnaire and PIOS).
  2. c) The full list of questions for the COVID-IO construct is provided on lines 334-352. I suggest just included a couple of examples in the Materials and Method section instead.

The full set of questions are provided as a key to the factor analysis output.

  1. Factor analysis seems to support the unidimensionality of the constructs COVID-TIME and COVID-IO. However, there is not really enough detail provided. I suggest checking the following paper by Slocum-Gori and Zumbo (2011): https://doi.org/10.1007/s11205-010-9682-8

We do not think that the factor analysis is a crucial part of the results. Many would just add up the time and information overload scores. We have now included the full output from the factor analyses in the supplementary material for those who might have a specific interest in the topic. The data is also available should anyone want to use other software or metrics to factor analyse the data.

  1. The description of the translation process has been improved, but there is still no details on where and how items were revised and evaluated in terms of linguistic or cultural equivalence. The explanation of this procedure is important for readers, particularly since you are providing the scale as supplementary material. However, the Chinese version should be included as well, so that it could be used by other researchers studying related constructs with Chinese populations.

The Chinese version is included in the supplementary material.

  1. There are still no sample items for PIOS in the manuscript.

Examples are now given in section 2.4.2 GENIO.

Reviewer follow-up:

  1. There is still no full correlation matrix. Table 3 only contains correlations between the predictor and outcome variables. A full matrix is preferred, particularly in establishing the rationale and meeting the assumptions for regression analysis.

Tables 3, 4 and 5 now give the complete set of correlations.

  1. No values for statistical significance are provided for the correlations.

Significance is shown under the Tables.

  1. As mentioned above, the discussion of FA and relevant statistics or assumption tests are not complete.

Complete output from the factor analyses is shown in the supplementary material.

Reviewer follow-up:

  1. a) It is nice to see the VIF statistics mentioned.
  2. b) I am confused by the mention of Change in F (line 395), as this seems to suggest that you compared the R2between the models for positive wellbeing as a DV and negative wellbeing as a DV. This is hard to understand.

The word “Change” is just default in SPSS. It is not appropriate when using the ENTER method so has been removed.

  1. Certainly it would make more sense to compare two models (one with the control variables and one without). In this context, the change in F would be a meaningful result that could be interpreted.

Additional analyses now involve the approach mentioned in point d.

  1. It is not convincing to state that “information overload reduces wellbeing and this is still observed when established predictors of wellbeing are controlled.” I think that you might have a contribution here, since the standardized coefficient and significance of the predictive power of information overload is demonstrated in the overall model (for negative wellbeing, particularly). However, the wording can be more precise and the argumentation and explanation more clear. Further analyses, such as comparison of high, medium, and low information overload participants in terms of other factors (such as through ANOVA) could provide better insights into the relationships among variables.

ANOVAS are now presented to confirm the regression analyses.

  1. There is no clear rationale for including the control variables (such as hypothesized effects on wellbeing), which should either be briefly discussed in the literature review or mentioned in the Methods or Results in terms of the regression analyses conducted.

The rationale for the inclusion of the established predictors is now given in section 1.3

Reviewer follow-up:

  1. This aspect has been improved in the Discussion and Conclusions. It would be a good idea to include some of that relevant discussion in the introduction, to contextualize your study. While these sources may have been published after you concluded your study, they can still be retrospectively adapted to establish a more clear research gap and rationale.

We think that inclusion of studies focusing on the nature of the information in the introduction would give the wrong impression about the aims of the present study.

  1. I respectfully disagree with the claim that “Most of these published studies have looked at some other aspect of information overload and not its association with wellbeing” (lines 587-588) as I have published on this precise topic.

“Most” has been replaced by “Some”.

  1. Again, a more nuanced approach (i.e., the nature of information which is being assessed for the construct of information overload) is strongly encouraged.

The discussion covers the nature of the information and we are clearly supporting this type of research. Indeed, in our research on internet addiction we have shown that time on the internet and the nature of the material are both important influences on wellbeing.

Reviewer follow-up: Citations 30 through 35 might not be absolutely necessary, unless the DRIVE model can be more clearly shown and explained (which was not done). That would justify the inclusion of several of Dr. Smith’s seminal works in the field.

These are now relevant to the presented DRIVE model.

Additional feedback

Please use more sub-headings (particularly related to your key research questions).

This has been done.

In addition to hypotheses, clear research questions in a separate sub-section at the end of the introduction would be helpful. This can help to provide a more clear framework for the writing of each section (consistently).

The aims and hypotheses are now included in section 1.4.

The supplemental material is quite long and does not clearly divide the questions into sections which can be used in reference to the instruments mentioned in the Methods section. Clearer formatting of this file, if it is to be included with your submission, is required.

We are depositing the data on ResearchGate and one possible solution to the length of the supplementary material might be to put the English and Chinese versions of the survey there as well. The supplementary file would then just consist of the full factor analyses and ANOVAS.

Reviewer 3 Report

That Authors have considered reviewers' comments and presented highly amended manuscript. The new version is much more informative and improved. As new data is added the Tables should have appropriate numbers (especially when referred to in the text).    

Author Response

Tables now have appropriate numbers.